# The Janus Face of Oxidative Stress and Hydrogen Sulfide: Insights into Neurodegenerative Disease Pathogenesis

**DOI:** 10.3390/antiox14030360

**Published:** 2025-03-19

**Authors:** Constantin Munteanu, Anca Irina Galaction, Gelu Onose, Marius Turnea, Mariana Rotariu

**Affiliations:** 1Department of Biomedical Sciences, Faculty of Medical Bioengineering, University of Medicine and Pharmacy “Grigore T. Popa”, 700454 Iasi, Romania; anca.galaction@umfiasi.ro (A.I.G.); mariana.rotariu@umfiasi.ro (M.R.); 2Neuromuscular Rehabilitation Clinic Division, Clinical Emergency Hospital “Bagdasar-Arseni”, 041915 Bucharest, Romania; gelu.onose@umfcd.ro; 3Faculty of Medicine, University of Medicine and Pharmacy “Carol Davila”, 020022 Bucharest, Romania

**Keywords:** oxidative stress, hydrogen sulfide (H_2_S), neurodegenerative diseases, redox signaling, Janus face, neuroinflammation

## Abstract

Oxidative stress plays an essential role in neurodegenerative pathophysiology, acting as both a critical signaling mediator and a driver of neuronal damage. Hydrogen sulfide (H_2_S), a versatile gasotransmitter, exhibits a similarly “Janus-faced” nature, acting as a potent antioxidant and cytoprotective molecule at physiological concentrations, but becoming detrimental when dysregulated. This review explores the dual roles of oxidative stress and H_2_S in normal cellular physiology and pathophysiology, focusing on neurodegenerative disease progression. We highlight potential therapeutic opportunities for targeting redox and sulfur-based signaling systems in neurodegenerative diseases by elucidating the intricate balance between these opposing forces.

## 1. Introduction

Neurodegenerative disorders, such as Alzheimer’s disease, Parkinson’s disease, and amyotrophic lateral sclerosis (ALS), are a growing medical challenge worldwide [1,2,3]. These conditions are typically characterized by progressive neuronal loss, accumulation of misfolded or aggregated proteins, peroxisomal dysfunction [4,5], and persistent neuroinflammation [6,7,8,9,10]. While each disorder has distinct clinical features—ranging from cognitive decline to motor dysfunction, there is an overarching theme that oxidative stress contributes significantly to the pathological cascades involved [11,12,13]. Excessive ROS in the central nervous system (CNS) can damage neuronal membranes through lipid peroxidation, alter synaptic function, and potentially accelerate protein misfolding [14,15,16,17].

Concurrently, the inflammatory microenvironment within the brain can exacerbate oxidative stress, creating a feed-forward loop of cellular injury [18,19]. Emerging data show that alterations in the gut microbiota can affect the integrity of the blood–brain barrier (BBB), modify the immune landscape, and alter levels of neuroactive metabolites, all of which can converge to amplify oxidative stress in the brain [20,21,22,23].

Under healthy conditions, controlled levels of ROS are generated as part of natural metabolic processes, playing indispensable roles in cell signaling, host defense, and maintaining homeostasis [24,25,26].

When ROS production exceeds the buffering capacity of endogenous antioxidant mechanisms, this state is characterized by the potential for damage to critical macromolecules, including nucleic acids, lipids, and proteins, thereby altering their structure and function [27]. In many diseases—ranging from metabolic syndromes to progressive neurodegenerative disorders—there is a growing consensus that oxidative stress acts as an aggravating factor that accelerates tissue dysfunction and cell death [28]. Moreover, oxidative stress can also act as a driver of inflammation and tissue degeneration, creating a self-perpetuating cycle that amplifies pathophysiological processes [29].

Despite decades of intense research into antioxidants, the multifaceted nature of oxidative stress regulation continues to confound simple therapeutic interventions [30]. This complexity arises partly because ROS are not universally deleterious; at moderate concentrations, they are essential for immune defense, redox signaling, and various physiological adaptations [31]. Consequently, outright suppression of ROS can be counterproductive, as it may disrupt beneficial signaling pathways [32]. Researchers now recognize the importance of striking a precise balance, where the cellular environment maintains sufficient oxidative potential for physiological signaling, but not so much as to cause sustained damage [33]. Against this backdrop, the concept of a “Janus face” emerges for oxidative stress, encapsulating the idea that ROS can be both detrimental and instrumental, depending on their concentration, cellular localization, and temporal dynamics [34].

Hydrogen sulfide (H_2_S) once occupied a less prominent place in biomedical research, mainly because it was considered primarily as a toxic gas with a notorious smell [35]. Over the past several decades, H_2_S has been reclassified as a vital endogenous signaling molecule, joining nitric oxide and carbon monoxide in the growing family of gaseous neurotransmitters and modulators [36,37]. Endogenously, H_2_S is produced via several enzymatic pathways, most notably those involving cystathionine β-synthase, cystathionine γ-lyase, and 3-mercaptopyruvate sulfurtransferase, which operate in a tissue-specific manner [38]. At physiological concentrations, H_2_S exerts a multitude of beneficial effects: it promotes vasodilation [39] by activating potassium channels in vascular smooth muscle, modulates neuronal transmission in the central and peripheral nervous systems, and confers cytoprotection through antioxidant and anti-inflammatory mechanisms [40].

Research suggests that appropriate levels of H_2_S preserve mitochondrial function, attenuate endoplasmic reticulum stress, and modulate autophagic pathways. These diverse functions underline the broad physiological relevance of H_2_S in organs as distinct as the brain, liver, heart, and gut [41]. Intriguingly, just as oxidative stress exhibits a dual nature, H_2_S is also recognized for its “Janus face” [42]. While moderate H_2_S levels are protective and can mitigate oxidative injury, excessive H_2_S exposure may evoke detrimental outcomes [43]. In specific contexts, high concentrations of H_2_S can inhibit cellular respiration by targeting cytochrome c oxidase in the mitochondrial electron transport chain, leading to energy deficits and potential cell death [44]. This dichotomy accentuates the complexity of regulating H_2_S homeostasis, which is intimately tied to sulfur-containing amino acid metabolism and influenced by factors such as diet, microbial composition in the gut, and the functional capacity of endogenous enzymes [45,46].

Central to the emerging field of oxidative stress and H_2_S signaling is the realization that neither of these pathways operates in isolation. Instead, they are intricately woven into a larger molecular tapestry that includes inflammatory signals, metabolic regulators, and, increasingly, the gut microbiota [47]. These insights reveal a dynamic equilibrium in which oxidative stress and H_2_S act not merely as isolated factors, but as interconnected players in a broader biological network. Understanding how to manipulate this network to favor beneficial outcomes is a significant challenge—and opportunity.

## 2. Oxidative Stress: A Janus-Faced Phenomenon

ROS are generated through multiple biochemical routes in mammalian systems, with the mitochondrial electron transport chain (ETC) and the family of NADPH oxidases (NOX) representing two principal sources [48].

Within mitochondria, ROS production is intrinsically linked to oxidative phosphorylation. As electrons traverse the ETC—from NADH or FADH_2_ through complexes I to IV—a tiny fraction may prematurely reduce molecular oxygen, yielding superoxide (O_2_•^−^) [49]. Although this superoxide is rapidly dismutated by mitochondrial superoxide dismutase (MnSOD), the process can still lead to significant levels of hydrogen peroxide (H_2_O_2_) and downstream radicals under high metabolic flux or impaired electron transport conditions [50]. Notably, complexes I and III are often pinpointed as the significant sites of electron “leakage”, but recent evidence suggests that subtle changes in membrane potential or cofactor availability at various complexes can also influence the rate of ROS generation [51]. Mitochondrial ROS can, in turn, modulate processes such as mitochondrial fission/fusion dynamics, mitophagy, and apoptosis, thereby integrating redox signals with metabolic homeostasis [52,53].

Beyond mitochondria, specialized enzymes known as NADPH oxidases are dedicated ROS generators found in diverse cell types, including phagocytes, endothelial cells, and neurons. Unlike mitochondria, where ROS production is a byproduct of aerobic respiration, NADPH oxidases purposefully catalyze electron transfer from NADPH to oxygen, forming superoxide as their primary product. This controlled ROS output plays critical roles in cellular defense—particularly in immune cells, where NOX2 aids in microbial killing—and in normal signaling pathways that regulate gene expression, cell proliferation, and tissue remodeling [54]. Dysregulation of NOX isoforms, however, has been involved in various pathological states, including hypertension (through NOX1 and NOX5 overactivity), atherosclerosis, and neurodegenerative processes. For instance, sustained overactivation of NOX2 in microglia or astrocytes can contribute to neuroinflammation and oxidative neuronal injury [55].

The mitochondrial ETC and NADPH oxidases provide distinct yet complementary routes for ROS generation. The interplay between these sources is highly context-dependent, shaped by factors such as cellular metabolic demand, oxygen tension, and the activity of endogenous antioxidant networks. Redox-sensitive transcription factors and signaling cascades—such as nuclear factor erythroid 2-related factor 2 (Nrf2) or hypoxia-inducible factor-1α (HIF-1α)—sense shifts in ROS levels and orchestrate adaptive responses, including upregulation of antioxidant enzymes and metabolic reprogramming [55]. These regulatory pathways exemplify the intricacy of ROS homeostasis: while low-to-moderate ROS concentrations can act as signaling intermediates, excessive or prolonged elevations trigger oxidative stress, macromolecular damage, and cell death. Understanding these biochemical underpinnings is pivotal for elucidating the “Janus face” of oxidative stress—wherein ROS are both indispensable for normal physiological functions and potentially deleterious when dysregulated [56,57,58].

A network of endogenous antioxidant systems maintains the intricate balance between generating and eliminating ROS. At the forefront are several key enzymes: superoxide dismutase (SOD), catalase, and glutathione peroxidase (GPx). Their coordinated actions mitigate oxidative stress by detoxifying primary and secondary ROS, preserving cellular integrity and function [59].

A pivotal aspect of the “Janus-faced” nature of oxidative stress lies in its capacity to modulate signal transduction pathways that govern essential cellular processes, such as proliferation, differentiation, and survival [60]. Among the most extensively studied redox-sensitive regulators is Nrf2, which orchestrates a broad antioxidant and cytoprotective gene program [61].

Under basal conditions, Nrf2 is sequestered in the cytoplasm by the Kelch-like ECH-associated protein 1 (Keap1), a cysteine-rich scaffold that targets Nrf2 for ubiquitin-mediated degradation [62]. However, when intracellular ROS or electrophiles modify critical cysteine residues in Keap1, its conformation changes, hindering its ability to tag Nrf2 for destruction [63]. This permits newly synthesized Nrf2 to accumulate, translocate to the nucleus, and dimerize with small Maf proteins. The resulting heterodimer then binds to antioxidant response elements (AREs) in the promoters of genes that encode phase II detoxification and antioxidant enzymes, including glutathione S-transferases, heme oxygenase-1 (HO-1), NAD(P)H dehydrogenase [quinone] 1 (NQO1), and the catalytic and modulatory subunits of glutamate-cysteine ligase (GCL) [64].

By elevating the transcription of these cytoprotective genes, the Nrf2 pathway acts as a central hub for cellular adaptation to both endogenous and exogenous stressors [65]. Beyond simply mitigating oxidative damage, Nrf2 and its downstream targets also influence immune modulation, mitochondrial function, and metabolic reprogramming [66]. This multifaceted role highlights how carefully calibrated increases in ROS can serve as a biological “alert signal”, triggering Nrf2 activation and fortifying the cell against subsequent insults, while prolonged or overwhelming oxidative challenges can exhaust these protective mechanisms and tip the balance toward cell injury or death [67].

Other transcription factors, such as nuclear factor kappa B (NF-κB) and activator protein-1 (AP-1), are also redox-sensitive, adding further complexity to the cellular response [68]. Yet, Nrf2 remains a prime example of how cells leverage redox stress as a cue for adaptive gene expression. Pharmacological agents that enhance Nrf2 activity have garnered significant interest in the context of diseases characterized by chronic oxidative stress, including neurodegenerative disorders, cardiovascular pathologies, and metabolic syndromes [69].

In the realm of intracellular communication, moderate elevations in ROS can activate redox-sensitive signaling cascades, influencing the activity of proteins and transcription factors that regulate cell proliferation, differentiation, and apoptosis [70]. For example, transient bursts of ROS may modulate protein kinases or phosphatases by oxidizing their cysteine residues, fine-tuning the phosphorylation status of key regulatory proteins. These reversible modifications highlight the nuanced nature of ROS, acting as both a “switch” and a modulator for numerous cell fate decisions [71].

Equally important is the role of low-level ROS in orchestrating immune defenses [72]. Phagocytic cells, such as neutrophils and macrophages, depend on controlled ROS production to neutralize invading pathogens [73]. Through the so-called “respiratory burst”, NADPH oxidase generates superoxide radicals, which are converted into potent antimicrobial species that help to degrade bacterial and fungal intruders [74]. In addition to direct microbicidal action, these reactive molecules can also propagate signals that coordinate inflammatory responses, influencing the recruitment and activation of other immune cells. Significantly, when kept within physiological bounds, these ROS-mediated processes contribute to an effective immune surveillance system without causing widespread collateral damage to host tissues [75].

Despite the essential signaling roles that ROS fulfill at low concentrations, excessive accumulation can overwhelm the cell’s antioxidant defenses and precipitate a range of cytotoxic events. This transition from controlled redox signaling to harmful oxidative stress is often referred to as the critical “threshold” beyond which ROS cease to be beneficial, and instead become deleterious (Figure 1). In practice, no absolute numerical value demarcates this boundary; rather, the tipping point varies across cell types, tissues, and physiological conditions. However, the consequences of exceeding it are well documented, and include damage to virtually all classes of macromolecules—most notably, lipids, proteins, and nucleic acids [76].

Lipid peroxidation represents a prime example of ROS-induced cellular damage. Polyunsaturated fatty acids in cell membranes are particularly vulnerable to oxidative assault, leading to the formation of peroxyl radicals and aldehydic byproducts like malondialdehyde (MDA) and 4-hydroxynonenal (4-HNE). These toxic species can further propagate membrane destabilization, diminishing membrane fluidity and impairing critical functions such as receptor signaling and ion transport. In parallel, protein oxidation can alter the conformation and functionality of diverse proteins—ranging from cytoskeletal components to enzymes and receptors. These modifications may render proteins susceptible to misfolding and aggregation, ultimately impairing cellular pathways involved in metabolism, trafficking, and signal transduction [77].

DNA is not exempt from these damaging effects. Oxidative lesions, such as 8-hydroxy-2′-deoxyguanosine (8-OHdG), can accumulate in genomic and mitochondrial DNA. If unrepaired, these lesions can interfere with normal replication and transcription, promoting mutagenesis and compromising genomic stability. Consequently, persistently high levels of ROS have been implicated in accelerated aging and in the etiology of various pathologies, including cancer, cardiovascular disease, and neurodegenerative disorders. Cells do possess dedicated repair systems—such as base excision repair and nucleotide excision repair—that can mitigate the impact of DNA oxidation, but these mechanisms can themselves become overwhelmed when ROS levels are chronically elevated [78,79].

In neurodegenerative conditions, chronic oxidative stress can hasten neuronal loss by aggravating protein misfolding, mitochondrial collapse, and pro-inflammatory cascades in the brain [80]. Similar processes contribute to the progression of cardiovascular diseases, where enhanced oxidative stress promotes endothelial dysfunction, lipid peroxidation of low-density lipoproteins, and vascular inflammation. Metabolic disorders, such as diabetes mellitus, also exhibit heightened oxidative burdens that accelerate complications including nephropathy and retinopathy [81].

## 3. Hydrogen Sulfide Signaling: Another Janus-Faced Regulator

Endogenous H_2_S is synthesized through a network of tightly regulated enzymatic pathways that collectively modulate sulfur metabolism. Three primary enzymes—cystathionine β-synthase (CBS), cystathionine γ-lyase (CSE), and 3-mercaptopyruvate sulfurtransferase (3-MST)—are the main drivers of H_2_S biogenesis in mammalian tissues. Although they share specific substrates and cofactors, each enzyme is characterized by distinct tissue distributions, regulatory mechanisms, and metabolic outputs [82].

CBS catalyzes the condensation of homocysteine and serine to form cystathionine in the trans-sulfuration pathway. Traditionally recognized for its role in synthesizing cysteine from homocysteine, CBS also generates H_2_S under certain conditions. In the central nervous system, CBS is notably abundant in astrocytes, where its activity influences synaptic transmission, astroglial–neuronal signaling, and local redox balance. Dysregulation of CBS has been seen in neurodegenerative states, highlighting its importance beyond sulfur amino acid metabolism [83].

CSE breaks down cystathionine into cysteine, ammonia, and α-ketobutyrate, and can liberate H_2_S from cysteine or homocysteine. In contrast to CBS, which is more prevalent in the brain and liver, CSE exhibits significant expression in cardiovascular tissues—particularly in vascular smooth muscle cells. CSE-derived H_2_S contributes to vasodilation by activating potassium channels, thus playing a key part in blood pressure regulation. Moreover, CSE activity in immune cells is essential for modulating inflammatory processes and overall redox homeostasis [84].

Distinct from the trans-sulfuration route, 3-MST operates within a cysteine catabolic branch, wherein 3-mercaptopyruvate, produced by cysteine aminotransferase (CAT), is converted to H_2_S. Although found in multiple tissues, 3-MST is prominently expressed in mitochondria-rich organs such as the brain and kidney. Its significance in neuronal contexts has gained increasing attention, as 3-MST-generated H_2_S can modulate neuronal excitability, protect against oxidative stress, and intersect with signaling pathways tied to synaptic plasticity [84].

Collectively, these enzymatic pathways endow the body with a robust capacity to fine-tune H_2_S production in response to physiological demands, dietary inputs, and pathological challenges. Notably, they do not operate in isolation, but can be influenced by factors such as cellular redox status, hormonal signals, and even gut microbial metabolites. By coordinating H_2_S output across different organs, CBS, CSE, and 3-MST help to maintain sulfur homeostasis and support the multifaceted roles of H_2_S in cellular signaling, vascular function, and neuroprotection [85].

H_2_S homeostasis is intricately tuned by tissue-specific expression and regulation of the enzymes responsible for its synthesis—CBS, CSE, and 3-MST. Each tissue displays unique enzyme abundances and activity profiles, reflective of its distinct functional requirements and metabolic landscapes. For instance, the brain and liver exhibit substantial CBS activity due to their active trans-sulfuration pathways, whereas vascular smooth muscle cells rely more heavily on CSE to modulate blood pressure through H_2_S-driven vasodilation. Similarly, tissues with high mitochondrial density, such as the kidney and certain neuronal populations, utilize 3-MST-generated H_2_S for local redox signaling and cytoprotection [86].

Beyond this inherent tissue specificity, dietary sulfur amino acids—principally cysteine and methionine—profoundly influence H_2_S production. Diets rich in these precursors can enhance substrate availability for CBS and CSE, potentially increasing basal H_2_S levels. However, chronic overconsumption or malabsorption of sulfur-containing nutrients may also provoke dysregulated H_2_S synthesis if homeostatic feedback loops become overtaxed. Moreover, gut microbial metabolites—such as those originating from dietary proteins—can modulate host trans-sulfuration pathways by altering substrate pools and influencing enzyme expression or activity [87].

Host genetics further refines H_2_S regulation. Polymorphisms within the genes encoding CBS, CSE, or 3-MST may lead to functional changes in enzyme kinetics, altering H_2_S output and susceptibility to disease. For example, genetic variations that compromise CBS function can predispose individuals to hyperhomocysteinemia, which, in turn, impacts H_2_S availability and redox balance [88]. Conversely, gain-of-function mutations might elevate local H_2_S levels, intensifying cytoprotective or pro-inflammatory responses, depending on the physiological context [89]. Epigenetic modifications—such as methylation states affecting CBS or CSE promoter regions—also influence enzyme expression, reflecting a dynamic interplay between genotype, environment, and tissue-specific demands. Taken together, these multilayered factors underscore the complexity of H_2_S regulation across different organ systems, and highlight the importance of nutrition and genetic background in shaping the body’s capacity to harness H_2_S for both normal physiology and disease mitigation [90,91].

H_2_S has gained recognition for its robust anti-inflammatory and vasodilatory properties, positioning it as a crucial mediator of vascular health and immune regulation. In the inflammatory context, H_2_S helps to restrain hyperactive immune responses by modulating key signaling cascades [92]. Specifically, it can attenuate the nuclear translocation of pro-inflammatory transcription factors, thereby reducing the expression of cytokines such as interleukin-1β, interleukin-6, and tumor necrosis factor-α [93]. H_2_S can interfere with the activation of immune cells, including neutrophils and macrophages, limiting the release of additional reactive species and inflammatory mediators. These actions create a more balanced immune environment, preventing chronic or excessive inflammation that could otherwise lead to tissue damage and the progression of various diseases [94].

In parallel, H_2_S exerts potent vasodilatory effects through multiple mechanisms. One prominent pathway involves activating ATP-sensitive potassium (K_ATP) channels in vascular smooth muscle cells, causing membrane hyperpolarization and subsequent vascular wall relaxation. H_2_S may also enhance the bioavailability of nitric oxide (NO)—another critical vasodilator—by mitigating oxidative stress and preserving endothelial function. These synergistic interactions contribute to the maintenance of appropriate vascular tone, supporting healthy blood pressure regulation and tissue perfusion [95].

Beyond these primary roles, H_2_S influences other molecular targets that further amplify its cytoprotective effects. For instance, by preserving mitochondrial health and mitigating oxidative stress, H_2_S indirectly sustains endothelial and immune cell function under metabolic and inflammatory stress conditions. These anti-inflammatory and vasodilatory attributes underscore H_2_S’s integral role in safeguarding cardiovascular health and regulating immune homeostasis. When present at physiological levels, H_2_S can thus serve as a powerful ally against chronic inflammation and vascular dysfunction—key drivers of a wide range of pathological states, including atherosclerosis, metabolic disorders, and neurodegenerative conditions [96].

H_2_S plays a pivotal role in preserving mitochondrial integrity and function, a feature that is integral to its cytoprotective repertoire. Within the mitochondria, H_2_S can act as an electron donor, thus helping to sustain membrane potential and promote efficient oxidative phosphorylation. By bolstering mitochondrial respiration, H_2_S counteracts the onset of mitochondrial dysfunction—one of the earliest and most pervasive contributors to tissue injury and degeneration. Additionally, H_2_S exerts anti-apoptotic effects by stabilizing the mitochondrial membrane, thereby reducing the leakage of cytochrome c into the cytosol. This stabilization diminishes the formation of apoptosomes and interrupts downstream cascades that would otherwise culminate in programmed cell death [97].

Beyond safeguarding mitochondrial energetics, H_2_S also influences autophagy, the intracellular degradation pathway that is critical for cellular homeostasis. Autophagy helps to eliminate damaged organelles and misfolded proteins, thereby preventing their toxic accumulation [98]. By modulating signaling pathways, such as those involving mammalian target of rapamycin (mTOR) or AMP-activated protein kinase (AMPK), H_2_S can fine-tune autophagy initiation in response to oxidative and metabolic stress. For instance, conditions that elevate H_2_S levels may induce or enhance autophagic flux, ensuring timely clearance of dysfunctional mitochondria and other cellular debris. Conversely, impaired H_2_S production could diminish autophagic capacity, leaving cells more susceptible to oxidative damage and contributing to disease progression [99].

This interplay between H_2_S-mediated mitochondrial protection and autophagic regulation underscores the multifaceted nature of H_2_S as a cytoprotectant. By concurrently supporting mitochondrial bioenergetics and orchestrating quality-control mechanisms, H_2_S fortifies cells against acute stressors and helps to maintain long-term viability [100]. Dysregulation of H_2_S synthesis—whether from enzymatic deficiencies, genetic variations, or environmental factors—can thus amplify vulnerabilities in tissues that heavily rely on robust mitochondrial function, such as the brain, liver, and heart [101].

H_2_S exerts significant antioxidant effects by direct and indirect mechanisms that help to mitigate ROS accumulation. From a chemical standpoint, H_2_S can react with certain radical species—such as hydroxyl radicals—to form less harmful or more easily managed byproducts, thereby preventing ongoing oxidative damage [102]. H_2_S is not generally considered a potent one-to-one radical scavenger, due to very low H_2_S concentrations [103]. Data suggest that H_2_S’s primary antioxidant or “redox-modulating” effects occur through the upregulation of endogenous defense systems (e.g., via Nrf2 activation, persulfidation of relevant enzymes) [104], indirect inhibitory effects on pro-oxidant enzymes (e.g., partial inhibition of NADPH oxidases) [105], and formation of sulfane sulfur species (polysulfides), which can more effectively quench radicals or modify thiols in proteins [106,107]. H_2_S also influences key antioxidant pathways at the transcriptional and post-translational levels. For instance, it has been shown to upregulate endogenous antioxidant enzymes by activating redox-sensitive transcription factors, including Nrf2. Through Nrf2, H_2_S indirectly enhances the expression of cytoprotective proteins like SOD, glutathione peroxidase (GPx), and heme oxygenase-1 (HO-1), further bolstering the cell’s defenses against ROS [108].

In parallel, H_2_S can sustain or replenish critical cellular thiols, notably glutathione (GSH). Maintaining a favorable redox environment prevents GSH depletion under oxidative stress conditions, and supports ongoing detoxification of peroxides and other reactive species. H_2_S may also mitigate the activity of pro-oxidant enzymes—for example, by modulating NADPH oxidase isoforms—thereby reducing superoxide generation at its source. These multifaceted protective actions collectively position H_2_S as a central regulator of redox balance. When properly regulated, its ROS-scavenging or -suppressing capacities shield cells from oxidative insults; however, dysregulated H_2_S production, whether overly abundant or insufficient, can tip the scales toward oxidative injury, reflecting the broader “Janus face” paradigm inherent to sulfur-based and redox signaling [109].

While low-to-moderate concentrations of H_2_S are lauded for their cytoprotective, anti-inflammatory, and antioxidant properties, excessive H_2_S can be distinctly harmful. One of the primary targets of high H_2_S levels is cytochrome c oxidase (complex IV) in the mitochondrial electron transport chain. Through interactions at the heme and copper active sites of cytochrome c oxidase, elevated H_2_S can effectively block the final step of oxidative phosphorylation, impeding the transfer of electrons to molecular oxygen. This inhibition leads to a rapid decline in ATP production, imposing metabolic stress on cells and potentially triggering cell death pathways [110,111].

Such mitochondrial toxicity is particularly relevant in tissues with high energy demands, including neuronal and cardiac tissues. Even transient episodes of excessive H_2_S in the nervous system may compromise neuronal viability by depleting ATP stores and impairing essential ionic gradients that underlie electrical excitability. Similarly, sudden surges in H_2_S in cardiac tissue can disrupt the energetic equilibrium required for coordinated contraction and relaxation cycles, intensifying the risk of arrhythmias and ischemic injury. Beyond these immediate energetic consequences, partial cytochrome c oxidase inhibition may amplify the formation of ROS upstream in the electron transport chain, precipitating further oxidative damage [112].

Several factors dictate whether H_2_S exerts a protective or toxic effect. These include the rates of H_2_S production and consumption (e.g., via mitochondrial sulfide quinone oxidoreductase), tissue-specific enzyme expression, and the availability of alternative electron donors or acceptors. Host genetics and environmental inputs can also modulate H_2_S metabolism, further shaping its physiological or pathological outcome. Consequently, maintaining regulatory balance is paramount: small, carefully managed increases in H_2_S support critical signaling and protective pathways, whereas unchecked elevations can compromise respiratory efficiency and cell survival. This delicate equilibrium underscores the broader “Janus face” paradigm of H_2_S signaling, wherein the same molecule that safeguards mitochondrial integrity at low doses can sabotage it under conditions of high, unregulated accumulation [113].

H_2_S becomes acutely toxic at atmospheric levels beyond 500 ppm, while cellular concentrations exceeding roughly 5–30 µM disrupt respiratory enzymes [44]. Historically, early methods (e.g., methylene blue) exaggerated H_2_S to 50–160 µM in the plasma or brain by acid-liberating bound sulfide (Table 1). However, newer high-sensitivity techniques (gas chromatography, polarographic sensors, and MBB derivatization) indicate lower physiological ranges: in humans, ~0.7–3 µM in plasma; in mice, around 10–15 nM. These discrepancies reflect varying sample protocols, complex sulfide-binding pools, and dynamic H_2_S turnover [114,115]. Consequently, caution is essential when measuring or interpreting H_2_S levels, given the technical issues above and their role in health and toxicity.

Determining the precise concentration range within which H_2_S transitions from a cytoprotective agent to a cytotoxic threat remains a complex challenge. This threshold varies considerably across tissues, reflecting their metabolic demands, oxygen availability, and specific enzyme profiles for both H_2_S biosynthesis and degradation. H_2_S supports mitochondrial respiration, bolsters antioxidant defenses, and modulates inflammation at lower concentrations—typically in the nanomolar-to-low-micromolar range. Within these levels, H_2_S can scavenge ROS, upregulate endogenous antioxidant enzymes, and facilitate crucial signaling pathways that maintain vascular tone and cellular homeostasis [123].

However, once tissue levels of H_2_S surpass this protective window, the risk of mitochondrial dysfunction escalates significantly. High-micromolar-to-millimolar concentrations of H_2_S can bind to cytochrome c oxidase, inhibiting the final step of oxidative phosphorylation and leading to ATP depletion. Under such conditions, cells are forced into an energy crisis, rendering them more susceptible to apoptosis or necrosis. These detrimental outcomes are most pronounced in tissues that rely heavily on aerobic metabolism, such as the brain, liver, and myocardium, where even transient overexposure can trigger irreversible injury [124].

The exact thresholds are not only tissue-dependent, but also influenced by individual genetic polymorphisms, coexisting pathologies, and exogenous factors, such as diet or pharmaceuticals. Moreover, dynamic regulatory mechanisms—encompassing enzymes like sulfide quinone oxidoreductase (SQR) that catabolize H_2_S—can shift the balance between protection and toxicity in real time. Ultimately, the dose–response curve for H_2_S underscores its “Janus-faced” nature (Figure 2): while moderate levels confer substantial cytoprotection, a steep rise beyond key regulatory thresholds can swiftly transform H_2_S from a benevolent signal into a potent mitochondrial toxin [116,125,126].

In addition to endogenous mechanisms for H_2_S production, external sources of this gasotransmitter can considerably impact physiological and pathological processes. Synthetic H_2_S donors and inhibitors have emerged as valuable tools for both experimental research and potential therapeutic applications. By precisely modulating H_2_S levels in targeted tissues, these agents help to clarify the boundaries between cytoprotection and cytotoxicity, as well as offering avenues for treating conditions linked to redox imbalance, inflammation, and vascular dysfunction [127].

Exogenous sources [128,129,130,131] and H_2_S-releasing compounds vary widely in their chemical structures and kinetics. Early-generation donors, such as sodium hydrosulfide (NaHS) and sodium sulfide (Na_2_S), release H_2_S almost immediately upon dissolution in aqueous media, producing robust but relatively short-lived surges in H_2_S concentration. While useful for acute experiments, these rapid-release agents can induce supraphysiological levels of H_2_S that obscure the molecule’s endogenous “steady-state” functionality [132]. To address this limitation, more advanced H_2_S donors have been developed, featuring slow-release or targeted-release profiles. For example, GYY4137 decomposes gradually, maintaining moderate H_2_S concentrations over longer periods and more closely mimicking physiological conditions. Similarly, novel organosulfur-based compounds, H_2_S-loaded nanoparticles, and hybrid drugs (e.g., hybrid NSAIDs appended with H_2_S-releasing moieties) aim to localize H_2_S delivery to specific tissues or disease sites. By leveraging such controlled-release strategies, researchers can investigate how incremental H_2_S supplementation influences mitochondrial function, inflammation, vascular tone, and neuroprotection, without crossing cytotoxic thresholds [133].

Agents that suppress H_2_S biosynthesis or activity also play an essential role in probing its pathophysiological effects. For instance, inhibitors of CBS or CSE help to delineate each enzyme’s contribution to tissue-specific H_2_S pools. By selectively limiting H_2_S production, researchers can explore whether elevated endogenous H_2_S underlies maladaptive responses—such as excessive vasodilation or metabolic disruption—in certain disease states. Likewise, inhibitors of mitochondrial sulfide oxidation enzymes (e.g., sulfide quinone oxidoreductase or SQR) provide insights into how impaired H_2_S catabolism might drive toxic accumulation of this gasotransmitter [122].

In conditions characterized by reduced endogenous H_2_S (e.g., ischemic heart disease, certain neurodegenerative disorders), controlled H_2_S donors may restore cytoprotective signaling and enhance tissue resilience. Conversely, in diseases linked to pathological H_2_S overproduction or impaired clearance (e.g., certain gastrointestinal pathologies, sepsis), inhibitors of H_2_S biosynthesis or catabolism could mitigate toxicity [134].

## 4. The Interplay Between Oxidative Stress and H_2_S

H_2_S can exert significant control over ROS generation by directly influencing the activity and expression of key ROS-producing enzymes, most notably, the family of NADPH oxidases (NOX). NADPH oxidase complexes are widely recognized as major contributors to both physiological and pathological ROS production. Comprised of membrane-bound and cytosolic subunits, these enzyme complexes catalyze the reduction of molecular oxygen to superoxide (O_2_•^−^), thereby initiating a cascade of subsequent ROS formation [135].

Emerging evidence suggests that H_2_S can attenuate NADPH oxidase activity via multiple mechanisms. One avenue involves the post-translational modification of critical cysteine residues within NOX subunits. H_2_S can form persulfide (-SSH) groups, resulting in conformational changes that diminish the enzyme’s catalytic efficiency. Persulfidation typically involves reactive sulfur intermediates formed enzymatically or via oxidation of H_2_S. The signaling mechanism often consists of those sulfane/sulfur modifications, rather than a simple direct H_2_S–thiol reaction. By altering subunit interactions or electron transfer dynamics, H_2_S effectively reduces the ability of NOX to generate superoxide anions. Additionally, H_2_S may suppress the expression of specific NOX isoforms at the transcriptional level, potentially by upregulating redox-sensitive transcription factors that favor an antioxidant gene profile (e.g., Nrf2) [119].

Beyond direct enzyme modulation, H_2_S can further limit excessive ROS production by promoting the activity or expression of endogenous antioxidants such as SOD and glutathione peroxidase (GPx). This indirect route lowers the cellular burden of superoxide and hydrogen peroxide, thereby creating a less favorable environment for oxidative stress to escalate. In the context of gut physiology, where NADPH oxidase-derived ROS can influence epithelial barrier function and microbial community composition, H_2_S’s regulatory actions may be particularly impactful. Moderate H_2_S levels help to maintain a balanced oxidative environment, supporting mucosal integrity and mitigating inflammation [136].

However, these beneficial effects hinge on maintaining H_2_S concentrations within a physiologically relevant range. Excessive H_2_S may drive other deleterious processes (such as cytochrome c oxidase inhibition), underscoring the delicate balance inherent to sulfur and redox homeostasis. In this manner, H_2_S’s modulatory influence over NADPH oxidase exemplifies the dynamic and sometimes paradoxical interplay among oxidative stress, H_2_S signaling, and the gut microbial milieu [137].

ROS can profoundly influence the catalytic activity of enzymes responsible for H_2_S production, including CBS, CSE, and 3-MST. These enzymes depend on reactive thiol residues and, in some cases, essential metal cofactors or pyridoxal phosphate (PLP). When ROS levels rise, the thiol groups of these enzymes can undergo oxidative modifications, such as sulfenylation (–SOH), sulfinylation (–SO_2_H), or sulfonylation (–SO_3_H). Although the initial oxidation to a sulfenic acid may be reversible, further oxidation can become irreversible, permanently impairing enzyme function. Alternatively, S-glutathionylation can occur when glutathione conjugates with oxidized cysteine residues, providing temporary protection, but simultaneously blocking the active site until the modifying group is removed. Oxidative stress can also disrupt metal cofactor states, weakening enzyme–substrate affinities or destabilizing protein structure. Over time, these modifications reduce H_2_S output by diminishing the enzymes’ catalytic efficiency and may hasten their degradation or misfolding [138].

A decrease in H_2_S levels has far-reaching consequences, because H_2_S typically plays a protective role in redox balance and mitochondrial integrity. Under moderate conditions, H_2_S helps to maintain the function of respiratory complexes, scavenges free radicals, and regulates inflammation by modulating key signaling pathways. However, when excessive ROS inhibit H_2_S-synthesizing enzymes, the diminished H_2_S supply undermines these protective functions, exacerbating oxidative stress and altering cellular homeostasis. This deficit in H_2_S can also influence the gut microbiota, which relies on both sulfur availability and host redox status to maintain a stable ecosystem [139].

Redox and sulfur-based signals converge on multiple intracellular pathways that dictate how cells adapt to or counteract oxidative stress. One key integrator is the Nrf2 system, which responds to changes in both ROS and H_2_S. When ROS levels rise, oxidative modifications to Keap1 lift the inhibitory hold on Nrf2, allowing it to translocate into the nucleus and induce genes related to antioxidant defense. H_2_S can enhance this response by forming persulfide bonds on critical cysteines in Keap1 or by acting as a mild electrophile, thus further promoting Nrf2 activity. Through such cooperative mechanisms, cells upregulate enzymes like glutathione peroxidase and SOD, collectively controlling harmful oxidation reactions [140].

In parallel, the AMP-activated protein kinase (AMPK) pathway senses shifts in the cellular energy landscape that often accompanies ROS generation and H_2_S fluctuations. When ATP levels drop under stress, AMPK activation can initiate protective processes, including autophagy and metabolic reprogramming. Both exogenous and endogenous H_2_S modulate AMPK signaling by maintaining mitochondrial efficiency and balancing redox cues. This tight interplay ensures that H_2_S availability informs the cell’s decision to ramp up energy conservation or accelerate the clearance of damaged organelles through autophagy [141,142].

Additional crosstalk emerges in inflammatory contexts, where nuclear factor kappa B (NF-κB) orchestrates immune-related gene expression. Low levels of H_2_S tend to dampen NF-κB activation by reducing oxidant-mediated cytokine release, while excessive H_2_S or overwhelming ROS can tip the balance toward chronic inflammation. Such regulatory loops highlight how redox modifications and sulfur-based signals intersect, influencing transcriptional networks that mediate inflammation, antioxidant defense, and metabolic recalibration. By integrating these convergent pathways, cells align their stress responses with broader physiological shifts, safeguarding critical processes like mitochondrial integrity and immune surveillance [143].

## 5. Implications for Neurodegenerative Disease Pathogenesis

Oxidative stress has emerged as a unifying feature of multiple neurodegenerative disorders, including Parkinson’s, Alzheimer’s, and Huntington’s diseases, under its potent capacity to disrupt protein homeostasis, compromise mitochondrial function, and induce neuronal death [144]. Chronic elevations in ROS can facilitate pathological protein aggregation, such as alpha-synuclein in Parkinson’s or amyloid-beta and tau in Alzheimer’s, by oxidizing key amino acid residues and destabilizing protein conformation. These aggregates can, in turn, compromise proteasomal and autophagic pathways, propagating a vicious cycle of misfolded protein accumulation [145]. Simultaneously, mitochondria—already vulnerable in the aging or diseased brain—become further compromised under oxidative stress, given that ROS can damage the electron transport chain, mitochondrial DNA, and membrane integrity, ultimately impairing ATP production and exacerbating energy deficits [146]. As neurons rely heavily on oxidative phosphorylation and have limited regenerative capacity, persistent oxidative insults tip them toward apoptosis or necrosis, manifesting as progressive neuronal loss [147]. These molecular events are frequently accompanied by heightened neuroinflammation, wherein activated microglia and astrocytes generate additional ROS and pro-inflammatory cytokines, aggravating oxidative and proteotoxic damage [148]. In this interconnected pathological landscape, oxidative stress emerges not simply as an epiphenomenon, but as a core driver of disease progression, amplifying the accumulation of misfolded proteins, hastening mitochondrial dysfunction, and culminating in the death of neurons that are critical for motor control, memory, and cognition [149].

One of the most challenging aspects of understanding oxidative stress is determining what constitutes an “optimal” level of ROS. While cells require a baseline level of ROS to support physiological processes—such as immune defense and redox-sensitive signaling—the exact concentration range that maintains homeostasis without inducing damage remains elusive [150]. This question becomes particularly complex when considering different tissues and disease states. For instance, skeletal muscle might tolerate higher ROS levels, due to its robust antioxidant capacity and reliance on regulated oxidative signals for contractile adaptation [151]. In contrast, neurons are generally more vulnerable to oxidative injury because of their high metabolic rate and relatively limited regenerative capacity, implying a narrower window between beneficial and deleterious ROS levels [152].

Additionally, the temporal component of ROS exposure complicates efforts to define optimality. Acute, transient increases in ROS can activate adaptive pathways that enhance cellular resilience, whereas prolonged elevations can overwhelm antioxidant defenses, leading to chronic damage. The interplay between diverse ROS—such as superoxide, hydrogen peroxide, and hydroxyl radicals—further adds layers of complexity, as each species has unique reactivities and half-lives. Likewise, inter-individual variability in genetics, diet, and microbiota composition can drastically influence baseline redox states, making a universal “optimal” level improbable.

A longstanding debate in redox biology concerns whether brief, moderate increases in ROS can indeed confer neuroprotection, or whether they invariably hasten neurotoxic outcomes. On the one hand, a growing body of research supports the concept of “hormesis”, whereby low-dose oxidative insults trigger adaptive responses that fortify neuronal defense systems. For example, transient elevations in ROS have been shown to activate redox-sensitive transcription factors—such as Nrf2—leading to upregulation of antioxidant and cytoprotective genes. This may help neurons to better withstand subsequent, more severe challenges. Furthermore, mild oxidative signals can modulate synaptic plasticity, influencing processes like long-term potentiation or depression, which are integral to learning and memory [153]. On the other hand, the narrow margin of safety in neuronal tissue complicates this narrative. The brain’s high metabolic demand and limited regenerative capacity can transform even short-lived ROS surges into deleterious events, particularly if endogenous scavenging systems are compromised or if the bursts occur repeatedly [154].

In pathological settings such as early-stage Alzheimer’s or Parkinson’s disease, neurons often exist on the cusp of redox imbalance, meaning that minor additional oxidative insults might tip cells into irreversible damage, rather than eliciting a protective response. Discrepancies in experimental outcomes also reflect differences in model systems, ROS detection methods, and the specific timing and localization of ROS generation. Thus, whether mild oxidative bursts ultimately serve as preconditioning stimuli or catalysts for neuronal injury remains an open question. Clarifying the precise cellular contexts, thresholds, and regulatory pathways that govern ROS-mediated neuroprotection versus neurotoxicity is essential for designing targeted interventions that exploit beneficial oxidative signaling while minimizing the risk of exacerbating neurodegenerative processes [155].

Preclinical investigations in various animal models have underscored the potential of H_2_S-modulating strategies to alter neurological outcomes, offering insights into both therapeutic and mechanistic dimensions [114]. In rodent models of Parkinson’s disease, administration of H_2_S donors has been reported to ameliorate motor deficits and mitigate dopaminergic neuronal loss, possibly by bolstering mitochondrial respiration, suppressing neuroinflammation, and counteracting oxidative stress [156]. Similarly, in lab models of Alzheimer’s disease, H_2_S supplementation has been shown to improve spatial learning and memory performance, coinciding with diminished amyloid-β deposition and attenuated microglial activation. These protective benefits have been attributed, in part, to H_2_S’s ability to preserve synaptic integrity and dampen pro-inflammatory cascades [157]. Conversely, the use of inhibitors targeting endogenous H_2_S-producing enzymes has illuminated a dualistic nature: in scenarios where H_2_S levels are pathologically elevated, inhibiting CBS or CSE can restore redox balance and prevent metabolic disruptions that exacerbate neuronal injury. However, in normative or low-H_2_S contexts, such inhibition may remove essential cytoprotective mechanisms, heightening vulnerability to oxidative stress [158,159].

Redox indices, circulating and tissue H_2_S concentrations, and specific microbial signatures have emerged as promising biomarkers for assessing disease risk and tracking progression in neurodegenerative disorders [160]. Elevated oxidative stress markers—including lipid peroxidation products (e.g., malondialdehyde), protein carbonyls, and oxidatively modified DNA bases—can signal heightened redox imbalance that often precedes or accompanies neuronal loss [161]. Simultaneously, atypical shifts in H_2_S levels—whether excessive or deficient—may serve as an early indicator of dysregulated sulfur metabolism and impending mitochondrial dysfunction. This imbalance in H_2_S homeostasis can also align with emerging microbial signatures identified through next-generation sequencing [118].

## 6. Current and Prospective Therapeutic Strategies

Antioxidant therapies, which aim to quell excessive ROS and mitigate oxidative damage, have long been explored as potential treatments for diverse conditions, including neurodegenerative diseases, cardiovascular disorders, and metabolic syndromes. Many such strategies target broad oxidative processes using exogenous antioxidants, such as vitamin E, vitamin C, polyphenols, or synthetic compounds like N-acetylcysteine. While beneficial in concept, these interventions have encountered notable challenges and limitations in clinical application [162].

One major issue lies in the non-specific nature of many antioxidant molecules, which indiscriminately scavenge ROS, regardless of their physiological or pathological origin. Because low or transient levels of ROS can be essential for signaling pathways—such as those involved in immune surveillance, cell proliferation, and redox-sensitive gene expression—blunting these signals can inadvertently suppress beneficial cellular responses [163]. Furthermore, the pharmacokinetics of systemic antioxidants can be highly variable. Some compounds display poor bioavailability or limited tissue penetration, whereas others may be rapidly metabolized or excreted before exerting therapeutic effects [164]. In neurodegeneration, it has proven challenging to deliver antioxidant agents across the BBB at concentrations sufficient to counteract the complex, multi-factorial oxidative insults within the CNS [165,166].

Additionally, the antioxidant “caging” of certain reactive intermediates may generate secondary byproducts with their own pro-oxidant or inflammatory potential, thereby diminishing therapeutic gains [167]. Clinical trials have often yielded inconclusive or modest results, emphasizing the difficulty of translating robust in vitro antioxidant capacity into meaningful in vivo outcomes for patients. Consequently, redox modulation strategies are evolving from broad ROS scavenging into more targeted approaches. These include site-specific or organelle-specific antioxidants, molecules designed to trigger endogenous antioxidant defenses (e.g., Nrf2 activators), and combination therapies that integrate antioxidants with anti-inflammatory or mitochondrial support agents [168]. By refining the specificity, delivery, and mechanistic alignment of antioxidant interventions, there is renewed promise for reducing the detrimental consequences of oxidative stress while preserving its essential physiological functions [169].

Targeted redox-based interventions aim to mitigate excessive oxidative damage without undermining the essential physiological roles of ROS [170]. Among the most promising strategies is the activation of Nrf2, a transcription factor central to the cellular antioxidant defense system. Nrf2 activators—such as sulforaphane, dimethyl fumarate, and certain synthetic compounds—induce a comprehensive cytoprotective program that upregulates glutathione biosynthesis, SOD, and other antioxidants or phase II detoxifying enzymes [171]. This endogenous amplification can fine-tune the redox balance more effectively than blanket ROS scavengers, reducing the likelihood of disrupting critical signaling pathways [172]. Nrf2 activation can dampen inflammation by curtailing the release of pro-inflammatory mediators, highlighting the multifaceted benefits of targeting this pathway in chronic oxidative stress conditions, including neurodegenerative diseases [173].

Mitochondria-specific antioxidants represent another refined approach for safeguarding cells against pathological ROS elevations. These agents—exemplified by MitoQ, SkQ1, and other triphenylphosphonium (TPP+) conjugates—are engineered to accumulate within the mitochondrial matrix, where a significant proportion of cellular ROS originates [174]. By preferentially scavenging superoxide and related species at their primary source, mitochondria-targeted antioxidants reduce the collateral damage to mitochondrial DNA, lipids, and proteins that can compromise cellular energy production. This localized action spares cytosolic or extracellular ROS-mediated signaling pathways, reducing the risk of dampening beneficial redox signals [175].

While challenges persist—such as optimizing dosing, ensuring stable accumulation in the mitochondria, and translating findings from animal models to human clinical practice—these emerging interventions illustrate the potential of precision-based redox management. By specifically modulating molecular triggers of oxidative imbalance, researchers and clinicians stand a better chance of controlling excessive oxidative stress without sacrificing ROS’s indispensable physiological roles [176,177].

H_2_S donors with controlled release profiles represent a significant advancement in therapeutic strategies targeting conditions characterized by oxidative stress, inflammation, and cellular dysfunction [178]. Unlike early-generation H_2_S donors, such as sodium sulfide (Na_2_S) or sodium hydrosulfide (NaHS), which release H_2_S in a rapid and non-specific manner, controlled-release donors are designed to deliver H_2_S gradually, mimicking endogenous production [179]. This approach ensures sustained therapeutic effects, such as cytochrome c oxidase inhibition or excessive vasodilation, with low toxicity risks. The slow and controlled release allows H_2_S to maintain physiological levels, preserving its cytoprotective and signaling functions without exceeding harmful thresholds [180].

Modern H_2_S donors are highly versatile, and include small-molecule compounds, stimuli-responsive systems, and advanced delivery platforms like nanoparticles and polymeric carriers [181,182]. Small-molecule donors, such as GYY4137, release H_2_S steadily over time, making them suitable for chronic conditions like neurodegenerative diseases and cardiovascular disorders [183]. On the other hand, stimuli-responsive donors release H_2_S in response to specific environmental triggers, such as pH changes or elevated ROS. For example, ROS-sensitive donors can precisely target oxidative stress hotspots, providing localized antioxidant effects in damaged tissues [115,184]. Nanoparticle-based and polymer-encapsulated donors enhance the specificity and bioavailability of H_2_S delivery, enabling targeted treatment of organs such as the brain or heart [117]. Depending on the therapeutic needs, these systems can be customized for sustained or burst release, and can incorporate surface modifications to improve cellular uptake or tissue targeting [185,186].

Hybrid H_2_S-releasing compounds, which combine H_2_S donors with other therapeutic agents, offer additional opportunities for dual-function interventions [187]. For instance, H_2_S-releasing nonsteroidal anti-inflammatory drugs (NSAIDs) provide anti-inflammatory effects while protecting the gastrointestinal lining from damage—a common side effect of traditional NSAIDs [188]. Similarly, H_2_S-releasing agents combined with mitochondrial-targeting compounds show promise for enhancing mitochondrial function and mitigating ROS-driven damage in diseases such as Parkinson’s and Alzheimer’s [189,190].

Despite the potential of controlled-release H_2_S donors, challenges remain in optimizing their pharmacokinetics, ensuring long-term safety, and tailoring treatments to individual patient needs. The variability in endogenous H_2_S metabolism across individuals highlights the importance of personalized medicine approaches [191,192].

Safety considerations are equally important, particularly given the dual “Janus-faced” nature of H_2_S as both a cytoprotective agent and a potential mitochondrial toxin at high concentrations. Excessive H_2_S can inhibit cytochrome c oxidase in the mitochondrial electron transport chain, leading to ATP depletion, oxidative stress, and cellular dysfunction. Thus, precise control over H_2_S release is necessary in order to avoid cytotoxicity while maintaining its beneficial effects on redox balance, mitochondrial health, and inflammation. The potential for off-target effects, such as vasodilation-induced hypotension or gastrointestinal irritation, must be carefully evaluated during preclinical and clinical testing [193,194].

H_2_S donors exhibit considerable potential for synergistic effects when combined with antioxidant or anti-inflammatory agents, offering a multifaceted approach to mitigating oxidative stress and inflammation in various disease contexts [195]. H_2_S possesses intrinsic antioxidant and anti-inflammatory properties, such as scavenging ROS, modulating redox-sensitive pathways (e.g., Nrf2 activation), and attenuating pro-inflammatory signaling cascades. These effects can be amplified when paired with complementary agents, enhancing therapeutic efficacy and broader coverage of pathological processes [120,121].

Synergy with antioxidant agents arises from the ability of H_2_S to enhance endogenous antioxidant defenses while directly neutralizing ROS. For instance, combining H_2_S donors with exogenous antioxidants like N-acetylcysteine (NAC), vitamin C, or coenzyme Q10 may simultaneously bolster glutathione synthesis, scavenge existing ROS, and protect mitochondrial integrity [196,197,198]. Moreover, H_2_S facilitates the activation of Nrf2, which governs the transcription of several key antioxidant enzymes, creating a positive feedback loop that amplifies the cell’s natural capacity to counteract oxidative damage. These combined effects may prove especially beneficial in diseases with overwhelming oxidative stress, such as neurodegenerative disorders, cardiovascular conditions, and ischemia–reperfusion injuries [199,200].

Similarly, the anti-inflammatory effects of H_2_S donors can complement traditional anti-inflammatory agents, including nonsteroidal anti-inflammatory drugs (NSAIDs), corticosteroids, or novel biologics targeting cytokine pathways. H_2_S has been shown to suppress the activation of nuclear factor kappa B (NF-κB), reduce the release of pro-inflammatory cytokines (e.g., IL-1β, TNF-α), and promote the resolution of inflammation by modulating macrophage phenotypes. When paired with NSAIDs, H_2_S donors can not only enhance their anti-inflammatory effects, but also mitigate NSAID-induced gastrointestinal toxicity by preserving mucosal integrity through enhanced vascularization and epithelial protection. Hybrid compounds, such as H_2_S-releasing NSAIDs, exemplify the therapeutic potential of such combinations [201,202,203].

Furthermore, H_2_S may work synergistically with agents targeting mitochondrial dysfunction, a common denominator in oxidative and inflammatory diseases. For example, combining H_2_S donors with mitochondria-specific antioxidants (e.g., MitoQ or SkQ1) could simultaneously protect mitochondrial DNA, proteins, and membranes from ROS-induced damage, while ensuring optimal electron transport chain function. These effects may reduce energy deficits and limit the activation of inflammatory cascades, creating a dual protective mechanism [204,205].

## 7. Controversies, Challenges, and Future Directions

Accurately measuring ROS and H_2_S levels in vivo presents significant methodological challenges, complicating efforts to understand their precise roles in physiological and pathological processes. Both ROS and H_2_S are highly reactive, short-lived molecules that exist at low concentrations in biological systems, making them difficult to quantify with precision. Their rapid turnover and localized signaling add further complexity, as these molecules often exert effects within specific microenvironments, such as mitochondria or cell membranes, rather than systemically [206,207].

For ROS, traditional detection methods such as fluorescence-based assays (e.g., dichlorodihydrofluorescein diacetate) and chemiluminescent probes often lack specificity, detecting a broad range of reactive species without distinguishing between key players like superoxide, hydrogen peroxide, and s. Furthermore, these probes are prone to auto-oxidation and may produce false positives under certain conditions, leading to overestimation of ROS levels. Advanced imaging techniques, such as two-photon fluorescence microscopy and electron paramagnetic resonance (EPR), offer higher specificity and spatial resolution, but are limited by their technical complexity, high cost, and inability to provide real-time, systemic ROS measurements [208,209].

Similarly, H_2_S quantification faces its own set of challenges. Standard approaches, such as colorimetric assays and gas chromatography, often measure total sulfide levels, rather than distinguishing between free H_2_S, bound sulfides, and other sulfur-containing species. This lack of specificity can obscure the biological relevance of H_2_S fluctuations. Furthermore, H_2_S is highly volatile, and its concentrations are easily influenced by sample handling and processing, which can lead to degradation or artificial generation of the molecule during analysis. Real-time in vivo detection is particularly challenging, although emerging methods, such as fluorescent probes (e.g., H_2_S-sensitive dyes) and electrochemical sensors, show promise. These technologies are still developing, and require refinement to improve their sensitivity, specificity, and compatibility with biological systems [210].

Another complication arises from the dynamic interplay between ROS and H_2_S, as these molecules can modulate each other’s levels through biochemical reactions, such as H_2_S scavenging of ROS or ROS-mediated inhibition of H_2_S-producing enzymes. This interdependence necessitates simultaneous, real-time measurement of both molecules in specific cellular or tissue contexts, which remains technically demanding. Moreover, the concentration thresholds for physiological versus pathological effects of ROS and H_2_S vary across tissues, species, and experimental conditions, making standardization difficult [211,212,213,214]. Addressing these methodological hurdles will require developing advanced, high-resolution techniques capable of detecting ROS and H_2_S in a spatially and temporally precise manner. Incorporating multi-modal approaches, such as integrating imaging with mass spectrometry or biosensors, could provide deeper insights into their localized dynamics and systemic roles. Overcoming these challenges is critical for elucidating the nuanced roles of ROS and H_2_S in health and disease, and for advancing the development of targeted redox and sulfur-based therapies [215,216].

Variability across experimental models and clinical studies remains a significant obstacle to understanding the precise roles of ROS, H_2_S, and gut dysbiosis in disease pathogenesis. Differences in animal models, study designs, and clinical parameters often lead to inconsistent or inconclusive findings, complicating the translation of experimental results into actionable clinical strategies [217].

## 8. Conclusions

The interplay between oxidative stress and H_2_S exemplifies the “Janus face” nature of biological signaling, where both can serve as either protective mediators or pathological drivers, depending on their concentrations, localization, and context. Maintaining an optimal redox balance and adequate levels of hydrogen sulfide (H_2_S) is crucial in order to protect against pathological cascades driven by oxidative stress in neuronal cells. While a basal amount of H_2_S supports antioxidant defenses, mitochondrial function, and controlled inflammatory responses, excessively high or insufficient H_2_S can disrupt homeostasis and promote neurodegenerative changes. Moreover, recent findings highlight that not only mitochondria, but also peroxisomes are profoundly affected by oxidative stress. Future investigations targeting both mitochondrial–peroxisomal integrity and fine-tuned H_2_S signaling may yield innovative therapeutic strategies for age-associated neurodegenerative diseases.

## Figures and Tables

**Figure 1 antioxidants-14-00360-f001:**
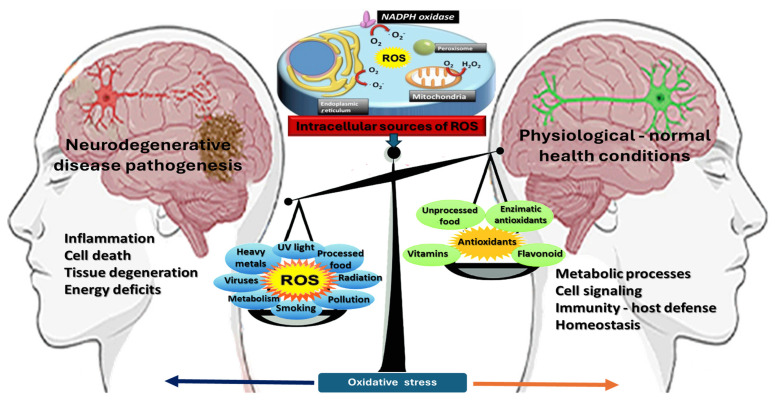
The Janus face of oxidative stress. This diagram illustrates the dual role of oxidative stress in cellular physiology and neurodegenerative disease pathogenesis, depicting a balance scale between physiological and pathological states. On the left, excessive ROS accumulation contributes to neurodegeneration through inflammation, neuronal apoptosis, mitochondrial dysfunction, and energy deficits, exacerbated by environmental and biological stressors such as heavy metals, radiation, smoking, and metabolic disorders. In contrast, the right side highlights the protective role of controlled ROS levels in cellular signaling, immune response, and mitochondrial homeostasis, supported by endogenous and dietary antioxidants. An inset diagram at the top-center illustrates ROS sources, including mitochondria, NADPH oxidase, peroxisomes, and the endoplasmic reticulum. This figure encapsulates the “Janus face” of oxidative stress, where moderate ROS levels maintain homeostasis, while excessive ROS drive neurodegeneration. The visual framework underscores the intricate interplay of oxidative stress, redox signaling, and neurodegenerative disease mechanisms.

**Figure 2 antioxidants-14-00360-f002:**
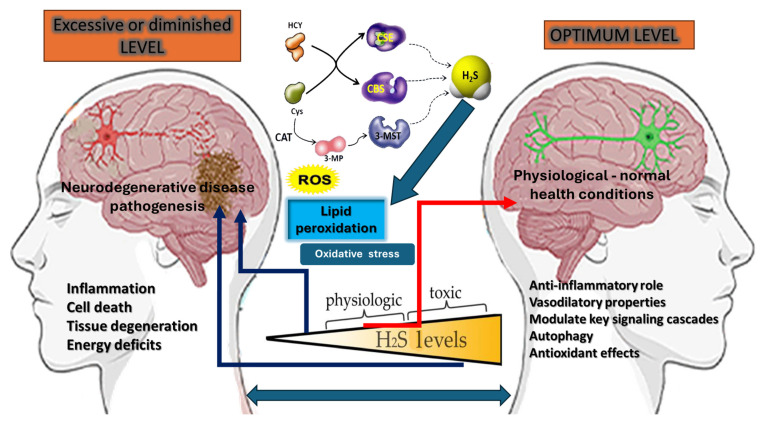
The “Janus face” of hydrogen sulfide (H_2_S) in neurodegenerative disease pathogenesis, emphasizing its dual role in physiological and pathological conditions. Excessive or diminished H_2_S levels, as shown on the left side, contribute to neurodegeneration by promoting inflammation, oxidative stress, lipid peroxidation, mitochondrial dysfunction, and energy deficits, which are similar to the effects of reactive oxygen species (ROS). This imbalance is depicted in a damaged brain with signs of inflammation, tissue degeneration, and neuronal cell death. Conversely, the right side represents optimal H_2_S levels that support physiological homeostasis by exerting anti-inflammatory effects, enhancing vascular function, modulating redox balance, and promoting neuroprotection. The figure also highlights key enzymatic sources of H_2_S—cystathionine beta-synthase (CBS), cystathionine gamma-lyase (CSE), and 3-mercaptopyruvate sulfurtransferase (3-MST)—which regulate H_2_S production. The interplay between ROS, oxidative stress, and lipid peroxidation underscores the necessity of maintaining balanced H_2_S levels to prevent neurodegeneration and support brain health, making H_2_S a crucial target for therapeutic interventions.

**Table 1 antioxidants-14-00360-t001:** Reported hydrogen sulfide (H_2_S) concentrations under different conditions.

Measurement Method	Sample/Tissue	Reported H_2_S Level	Observations/Comments	References
Methylene blue (strong-acid-liberation)	Rodent brain slices	50–160 µM	Potential overestimation due to the acid-labile release of sulfane sulfur pools.	[116]
Methylene blue + acid reagents	Mammalian plasma or tissue	>30–35 µM	Early measurements. Possibly includes “bound” or polysulfide forms. Generally recognized as artificially high.	[114]
Methylene blue	H2S in blood	2–5 μM	The human colon has the highest luminal concentration of H_2_S in the body (1.0–3.4 mM).	[117]
Gas chromatography (GC)	Rodent tissues	Typically nM to <1 µM	Researchers demonstrated that older, much higher (µM–100 µM) measurements were “dramatically overestimated”. The GC technique helps to avoid artifacts from strong acid steps.	[81,118]
GC + chemiluminescence	Blood (mouse)	~15–50 nM (plasma)	More realistic results for free (unbound) H_2_S in vivo.	[114,119]
Polarographic electrode	Various tissues	10–300 nM	A polarographic sensor in a sealed system. Helps to avoid volatility losses and acid-liberation artifacts.	[110,114]
Polarographic electrode	Blood and tissues (rodents)	50–200 nM	Researchers concur that free H_2_S is likely in the tens–hundreds of nM range, at least under baseline physiological conditions.	[114]
MBB (monobromobimane) derivatization + HPLC	Human liver or plasma samples	~0.7–3 µM	Less overestimation than with strong acid, but partial artifactual release can still occur, depending on pH and sample handling.	[116]
enzymatic assays, MBB/HPLC, and others	Human plasma	~0.4–1 µM	Suggests that the submicromolar–micromolar range for free H_2_S might be physiologically relevant in humans.	[120]
GC/polarographic/MBB	Vascular tissues, plasma (rodents)	Typically 100–300 nM	Repeatedly confirms baseline H_2_S in the ~nM-to-<1 µM bracket in healthy conditions.	[93,120]
Various (electrode-based, GC, fluorescent/bimane-based methods)	Cultured cells, animal tissues (brain, kidney, heart, retina)	nM–~1 µM	There is a broader consensus that “physiological” free H_2_S rarely exceeds a few µM. Levels above 10–50 µM appear toxic (rapid inhibition of cytochrome c oxidase, cellular respiration).	[112,115,121,122]

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
