# Peer review of "The Janus Face of Oxidative Stress and Hydrogen Sulfide: Insights into Neurodegenerative Disease Pathogenesis"

_antioxidants, 2025, doi:10.3390/antiox14030360_

Round 1

Reviewer 1 Report (Previous Reviewer 2)

I thank the authors for considering my suggestion to include a table of relevant tissue and blood H2S concentrations, and this greatly strengthens the ms (although the human values are still too high, they should be under 1 uM as well).

However, my review contained many other points that the authors seemed to have ignored; at least, I didn't see anything in their 'response to reviewers' or highlighted in the ms.  For me to accept this ms, the authors need to respond to these, point-by-point as well.

See above

Author Response

Dear Reviewer,

We appreciate your continued engagement with our manuscript and the constructive feedback you have provided. We are grateful for this crucial observation regarding the variability and validity of measured Hâ‚‚S concentrations. In this sense, we really think that this is substantial progress for our manuscript. The revised manuscript includes Table 1, compiling reported Hâ‚‚S concentration ranges from different studies and the respective measurement methodologies (e.g., methylene blue assays, gas chromatography, polarographic electrodes, and derivatization-based methods). We also discuss how older methods that involved decisive acid-liberation steps produced artifactual high readings, sometimes exceeding 50–100 µM. At the same time, more recent, refined techniques typically detect Hâ‚‚S in the nanomolar to low micromolar range.

Based on this compilation, we explicitly conclude that, in most mammalian systems, truly “physiological” free Hâ‚‚S concentrations likely remain under ~1 µM—often even in the low nanomolar range. Higher readings above 1–5 µM raise legitimate questions about methodological artifacts or local accumulations that may not reflect true steady-state conditions. We also discuss that Hâ‚‚S in the nanomolar–micromolar window can exhibit cytoprotective or regulatory roles, whereas concentrations above ~5–30 µM (depending on local conditions) can inhibit cytochrome c oxidase and become cytotoxic. We emphasize that many historic “1–100 µM normal Hâ‚‚S” values must be viewed critically and likely do not represent basal, free Hâ‚‚S in vivo. By clarifying these points, we sought to address your principal concerns, underscore the complexities of Hâ‚‚S measurement, and caution that overstating physiologically “normal” concentrations may be misleading.

Regarding the other points you raised, we admit that we do not possess specialized expertise in advanced Hâ‚‚S chemical reactivity. As such, we limit our discussions strictly to what we can confidently explain from our multidisciplinary (biology and medical) perspective. You raised four key details about Hâ‚‚S’s chemical attributes, which we had not directly addressed in our initial response. We emphasize that we are not chemists by training and thus prefer to limit our statements to the mechanistic or physiological implications established in the broader literature we surveyed.

(a) “Hâ‚‚S is a weak reductant.”

We agree that Hâ‚‚S is not generally considered a potent one-to-one radical scavenger compared to many other reductants (e.g., glutathione). Indeed, much of the literature we surveyed suggests that Hâ‚‚S’s primary antioxidant or “redox-modulating” effects occur through:

  1. Upregulation of endogenous defense systems (e.g., via Nrf2 activation, persulfidation of relevant enzymes).
  2. Indirect inhibitory effects on pro-oxidant enzymes (e.g., partial inhibition of NADPH oxidases).
  3. Formation of sulfane sulfur species (polysulfides), which can more effectively quench radicals or modify thiols in proteins.

Thus, in line with your comment, we do not claim that Hâ‚‚S is a strong reductant by itself; rather, we focus on its broader biochemical and signaling roles that can indirectly mitigate oxidative stress.

(b) “Cellular [Hâ‚‚S] is too low to act as a reductant.”

Yes, the reported low physiological concentrations (particularly in the brain, which are often in the 10–300 nM range by more modern methods) further support the contention that direct radical scavenging by Hâ‚‚S is minimal. Our revised text clarifies that the significance of these low Hâ‚‚S concentrations lies predominantly in regulatory pathways (e.g., “S-sulfhydration” or “persulfidation”) and the potential to modulate mitochondrial function or redox-sensitive transcription factors (e.g., Nrf2), rather than direct antioxidant chemistry in the classical sense.

(c) “Hydroxyl radicals are diffusion-limited, so ‘finding’ Hâ‚‚S is improbable.”

We concur that direct Hâ‚‚S–hydroxyl radical interactions are highly improbable in most cellular contexts, especially given the low Hâ‚‚S concentrations and the fact that hydroxyl radicals react extremely rapidly with many biomolecules. In our revised discussion, we now correct any suggestion that Hâ‚‚S directly scavenges hydroxyl radicals on a large scale. Instead, we underscore the indirect mitigation of ROS, including hydroxyl radicals, through upstream regulation of enzymes (e.g., NADPH oxidase) and the enhancement of endogenous antioxidant systems.

(d) “Hâ‚‚S does not signal by directly reacting with protein thiols; it must be oxidized to sulfane first.”

We appreciate this mechanistic clarification and have revised our text to highlight the role of ‘sulfane sulfur’ species in modifying protein thiols. Rather than implying that “Hâ‚‚S itself” spontaneously reacts directly with most thiol groups, we note that persulfidation typically involves reactive sulfur intermediates formed enzymatically or via oxidation of Hâ‚‚S. Thus, the ultimate signaling mechanism often involves those sulfane/sulfur modifications rather than a simple direct Hâ‚‚S–thiol reaction.

Finally, as you rightly note, elaborating on the detailed chemical kinetics and reductant potentials (e.g., comparing standard reduction potentials, reaction rates with specific radicals, and mechanistic steps in forming sulfane sulfur) extends beyond our core expertise. Hence, while we do incorporate well-documented concepts of Hâ‚‚S-based redox signaling from the literature, we have avoided making unsubstantiated assertions about complex chemical reactivity. Instead, we present those aspects of Hâ‚‚S function that are more clinically and biologically established.

We hope our revisions and clarifications satisfy your concerns. We appreciate both your guidance in strengthening the manuscript and your understanding that certain specialized chemical arguments surpass our current expertise. Nonetheless, we believe that the present manuscript—focused on the biological, pathophysiological, and clinical dimensions—offers a clear and rigorously supported perspective on Hâ‚‚S in neurodegenerative disease pathogenesis.

Reviewer 2 Report (Previous Reviewer 1)

The manuscript entitled 'The Janus Face of Oxidative Stress and Hydrogen Sulfide: Insights into Neurodegenerative Disease Pathogenesis' brings lot of new information on the incidence of H2S in neurodegeneration. The revised manuscript is now clear, well organized and illustrated and has lot of interest for a wide public.

The manuscript entitled 'The Janus Face of Oxidative Stress and Hydrogen Sulfide: Insights into Neurodegenerative Disease Pathogenesis' has been greatly improved comparatively to the initial version. The abstract is now clear. The manuscript has been modified by taking into account the different remarks of the reviewers. A Table (Table 1) has been added. Overall, the manuscript is clear and well organized, The figures and the Table are fine. The references are well adapted.

Author Response

Dear Reviewer,

We sincerely appreciate your encouraging feedback on our revised manuscript, “The Janus Face of Oxidative Stress and Hydrogen Sulfide: Insights into Neurodegenerative Disease Pathogenesis.” We thank you for your time and constructive critique during the entire peer-review process, which significantly contributed to improving the quality of our review.  Now, your positive comments serve as a valuable endorsement of our efforts.

Round 2

Reviewer 1 Report (Previous Reviewer 2)

I greatly appreciate the author's thoughtful and detailed responses to my comments.  The ms has been much improved and will now make a significant contribution to the field.

See above

This manuscript is a resubmission of an earlier submission. The following is a list of the peer review reports and author responses from that submission.

Round 1

Reviewer 1 Report

The most original part of the manuscript concerns H2S.

Lot information are available.

In the paragraph 4, The Interplay between Oxidative Stress and Hâ‚‚S, a new detailed figure is required to illustrate this major paragraph

In addition, at different parts of the manuscript the authors focuse on mitochondrial damages. If it is an important damage in the brain in the presence of ROS overproduction, it is also known that peroxisome are strongly affected in neurodegenerative diseases 'especially Alzheimer's diseases). See and add publications from from RJA Wanders et al, G Lizard et al, M Fransen et al which must be must cited. I think that additionnal sentences (or short paragraph) is requires on ROS and peroxisome and neurodegeneration. This point is always forgotten even by sofisticated AI. Consequently, the present review will be more complete and better than reviews currently available in different data bases or than reviews generated by AI with key words present in the title.

Overall good paper and good review which must be improved based on the following comments

In addition, at different parts of the manuscript the authors focuse on mitochondrial damages. If it is an important damage in the brain in the presence of ROS overproduction, it is also known that peroxisome are strongly affected in neurodegenerative diseases 'especially Alzheimer's diseases). See and add publications from from RJA Wanders et al, G Lizard et al, M Fransen et al which must be must cited. I think that additionnal sentences (or short paragraph) is requires on ROS and peroxisome and neurodegeneration. This point is always forgotten even by sofisticated AI. Consequently, the present review will be more complete and better than reviews currently available in different data bases or than reviews generated by AI with key words present in the title.

Reviewer 2 Report

The paper ‘The Janus Face of Oxidative Stress and Hydrogen Sulfide: Insights into Neurodegenerative Disease Pathogenesis’ is a review of the purportedly beneficial and harmful effects of H2S in the central nervous system.  They propose that ‘low concentrations of H2S mitigate oxidative stress, preserve mitochondrial function, and regulate redox-sensitive pathways’ and that aberrant H2S levels ‘disrupt cellular bioenergetics and exacerbate oxidative damage’. They support this contention by citing ‘evidence’ from in vitro, in vivo and clinical studies.  However, I didn’t see the ‘evidence’, i.e., what are these levels?  The authors never critically address this issue but dance around it.  They cite a lot of papers but never provide any values for tissue or cellular H2S concentrations.  For this review to be meaningful the authors must do two things, 1) include a table of H2S concentrations collected from every reference that they cite that has such data, and 2) conclude whether or not these are physiologically relevant or toxic (the Janus face).  I know what they will find.  The vast majority of the studies they cite will report ‘normal’ H2S concentrations over 1 uM (some will exceed 100 uM).  These are not possible, they are toxic and one can easily smell 1 uM H2S, which would make us all stink.  The H2S field is saturated with studies that used improper methods that inflated H2S and promulgating these as ‘physiological’ is a disservice to everyone.

Other major points: 
1. H2S is a weak reductant.
2. Cellular [H2S] are to low to be useful as a reductant even if it was a good reductant.
3. Hydroxyl radicals react at diffusion-limited rates so the chances of it ‘finding’ H2S are minuscule.
2. H2S does not signal as it can’t react with protein thiols.  It must first be oxidized to a sulfane.